# Filamentous Fungi Associated with Disease Symptoms in Non-Native Giant Sequoia (*Sequoiadendron giganteum*) in Germany—A Gateway for Alien Fungal Pathogens?

**DOI:** 10.3390/pathogens13090715

**Published:** 2024-08-23

**Authors:** Gitta Jutta Langer, Steffen Bien, Johanna Bußkamp

**Affiliations:** Department of Forest Protection, Northwest German Forest Research Institute (NW-FVA), Grätzelstraße 2, D37079 Goettingen, Germany; steffen.bien@nw-fva.de (S.B.); johanna.busskamp@nw-fva.de (J.B.)

**Keywords:** giant sequoia decline, causal factors and agents, invasive fungal species, fungal community, cases of disease, German forests, culture-based identification

## Abstract

Filamentous fungi associated with disease symptoms in non-native giant sequoia (*Sequoiadendron giganteum*) in Germany were investigated in ten cases of disease in Northwest Germany. During the study period from 2018 to 2023, a total of 81 species of Dikaria were isolated from woody tissue and needles of giant sequoia and morphotyped. Morphotypes were assigned to species designations based on ITS-sequence comparison and, in part, multi-locus phylogenetic analyses. Nine species were recognised as new reports for Germany or on giant sequoia: *Amycosphaerella africana*, *Botryosphaeria parva*, *Coniochaeta acaciae*, *C. velutina*, *Muriformistrickeria rubi*, *Pestalotiopsis australis*, *P. monochaeta*, *Phacidiopycnis washingtonensis*, and *Rhizosphaera minteri*. The threat posed to giant sequoia and other forest trees in Germany by certain, especially newly reported, fungal species is being discussed. The detection of a considerable number of new fungal records in the trees studied suggests that giant sequoia cultivation may be a gateway for alien fungal species in Germany.

## 1. Introduction

In the face of ongoing climate change, many European tree species are experiencing a loss of vitality and dieback, especially since the repeated drought and heat years from 2018 onwards. Prolonged periods of unusually high temperatures, high solar radiation, and precipitation deficits have led, for example, to the outbreak of complex disease and the vitality loss of beech in European beech (*Fagus sylvatica* L.) [1,2] in Germany. Native latent pathogens of beech, such as *Neonectria coccinea* (Pers.) Rossman & Samuels or *Biscogniauxia nummularia* (Bull.) Kuntze, have played a major role in the course of damage as key organisms. In detailed investigations of the fungal pathogens involved, *Diplodia corticola* A.J.L. Phillips, A. Alves & J. Luque was detected for the first time in Germany and worldwide on European beech [1,2]. So far, it has not been possible to clarify whether this species is native or whether it has been converted in the course of climate change and globalisation. *Diplodia corticola* is a latent plant pathogen that has frequently been isolated from physiologically impaired oak trees [3] and grapevine cankers [4].

Additionally, non-native tree species such as Douglas fir (*Pseudotsuga menziesii* (Mirbel) Franco) have come under drought stress under the environmental conditions since 2018, showing a loss of vitality and signs of death often associated with fungal pathogens [5]. In Germany, the main mortality factors of Douglas fir are fungi that have spread from native tree species to Douglas fir, such as *Armillaria ostoyae* (Romagn.) Herink, *Diplodia sapinea* (Fr.) Fuckel, *Heterobasidion annosum* (Fr.) Bref., and *Sirococcus conigenus* (Pers.) P.F. Cannon & Minter. Additionally, Douglas-fir-specific alien fungal species, such as *Nothophaeocryptopus gaeumannii* (T. Rohde) Videira, C. Nakash., U. Braun & Crous lead to a loss of vitality in the affected trees [5]. Besides climate change and globalisation, alien pathogens are among the most important triggers of emerging fungal diseases in forest trees [6].

Alien invasive organisms can have substantial negative impacts on forest ecosystems, their biodiversity, their species, and their communities [7,8,9]. Invasions by alien pests and pathogens can cause enormous damage to forests, leading to the near extinction of some tree species [9,10], as occurred, for example, with the introduction of the ash dieback pathogen *Hymenoscyphus fraxineus* (T. Kowalski) Baral, Queloz & Hosoya to Europe [11]. Invasions by non-native species are the fourth most important pressure directly driving global biodiversity loss [12]. Over the last 200 years, Europe has experienced an unprecedented increase in the number of forest pathogen introductions. This is largely due to human-assisted international transport and the trade of plants and plant materials [13,14,15]. As with plants, emerging infectious diseases of trees are closely linked to biological invasions and are often caused by the arrival of previously unrecognised pathogens or newly evolved species [16,17,18]. In this context, fungal and fungus-like infections have always played a major role [13,19]. The inventory of invasive forest pathogens (IFPs) in Europe, compiled by Santini et al. (2012) [13], has recorded 123 taxa (70% Ascomycota, Basidiomycota 21%, and 9% Oomycota), of which 42% are considered alien. For 28% of the IFPs, a European origin could be determined, but for 26% of species the actual origin was unclear. In almost 25% of all disease cases, IFPs lead to the death of the host tree. Commonly caused symptoms are tree dieback (37%) and growth reduction in the host tree (40%), or the death of the host tree (25%).

The giant sequoia (*Sequoiadendron giganteum* (Lindl.) J.Buchholz) in the family of *Cupressaceae* originates from California (Sierra Nevada). It grows primarily in the temperate biome and has been introduced to Austria, France, and Great Britain [20]. According to the German Federal Agency for Nature Conservation (BFN), there is an unstable neophytic occurrence in Germany. The giant sequoia was introduced to Europe in the mid-19th century and has since proved to be a largely hardy park tree [21]. As a light-demanding tree species, giant sequoia does not tolerate overshadowing or lateral pressure by other trees, which affects its ability to mix with other tree species [22]. When young, giant sequoia is sensitive to frost. In a growing trial in Germany, trees in some areas suffered from *Armillaria* [23].

Shoot dieback and Botryosphaeria canker on giant sequoia caused by *Botryosphaeria dothidea* (Moug.) Ces. & De Not. occur in both Europe [24,25,26] and North America [27]. The diseases are more pronounced in regions where giant sequoia is non-native [28]. *Botryosphaeria dothidea* (anamorph: *Fusicoccum aesculi* Corda) is the type species of the genus *Botryosphaeria* (*Botryosphaeriaceae*, *Botryosphaeriales*), but Slippers et al. 2004 [29] have shown that many fungi described under *B. dothidea* belong to other species. In the last 20 years, it has become apparent that *B. dothidea* is one of the most widespread and important endophytes, and this pathogen was found on a wide range agricultural and forestry plants [30]. *Botryosphaeria dothidea* is a host-stress-associated pathogen, and the stress is usually triggered by abiotic factors such as a lack of water or heat. Disease symptoms are cankers on twigs, branches and stems; the dieback of tips and branches; fruit rot; and blue stain [30]. Population analyses from Slovenia and Italy on European hop hornbeam (*Ostrya carpinifolia* Scop.) suggest that *B. dothidea* may be a native pathogen to Europe [31]. Additional *Botryosphaeriaceae* have been detected as pathogens on *S. giganteum*, such as *Botryosphaeria parva* (Pennycook & Samuels) Crous, Slippers & A.J.L. Phillips (anamorph: *Neofusicoccum parvum* (Pennycook & Samuels) Crous, Slippers & A.J.L. Phillips) in Switzerland [28] and *Neofusicoccum yunnanense* G.Q. Li & S.F. Chen in Croatia [32].

Fungi of the genus *Pestalotiopsis,* in particular *Pestalotiopsis funerea* (Desm.) Steyaert (*Pestalotiopsidaceae*), have also been implicated in disease outbreaks. The latter species is considered to be a weak secondary pathogen [25], to be widespread, and to cause leaf and stem blight, occasionally canker or girdling, and dieback and root rot, mainly on *Coniferae*, including giant sequoia [33,34].

The introduction of new tree species is also an important driver of new tree diseases. As non-native species can bring their pathogens with them from their country of origin, these alien species pose a potential threat to native tree species in Germany. Therefore, ten cases of disease in giant sequoia in Northwest Germany were analysed in detail with regard to their triggering and damaging factors in line with Manion (1981) [35] and Sinclair & Lyon (2005) [36]. The main objectives were as follows:To isolate and identify the filamentous Dikaria D. S. Hibbett, T. Y. James & Vilgalys species associated with the disease symptoms;To determine whether alien invasive fungal species were involved in the studied cases of disease;To discuss the risk to native tree species from alien fungal pathogens on giant sequoia in Germany.

## 2. Materials and Methods

### 2.1. Cases of Disease in Giant Sequoia in Northwest Germany

This study analysed only samples of giant sequoia, which were sent to the mycology laboratory of the Northwest German Forest Research Institute (NW-FVA) for a causal analysis of forest damage. Damages to giant sequoia trees in forest stands located in the federal states of Hesse and Lower Saxony were reported and investigated between 2018 and 2023. The samples of ten cases of disease observed in Northwest Germany (Table 1 and Appendix A, Figure 1) were analysed for associated fungi, particularly in transition zones from living to diseased or necrotic tissues, to identify the potential causal agent. However, in most cases, only twig samples (disease cases 1–6, 8, 10) were sent, and therefore no stem or branch tissue could be analysed. Detailed information on the diseased forest stands and sites as well as precipitation data are provided in the Appendix A. All of the studied sites had soils that were at least weakly mesotrophic or well supplied with nutrients, except for the forest sites of the disease cases 5 and 6. The soils in the latter were only weakly supplied with nutrients. The soils were mostly moderately fresh to fresh, in some cases, moderately dry in the summer, and weakly alternatingly moist to waterlogged or stock fresh.

**Figure 1 pathogens-13-00715-f001:**
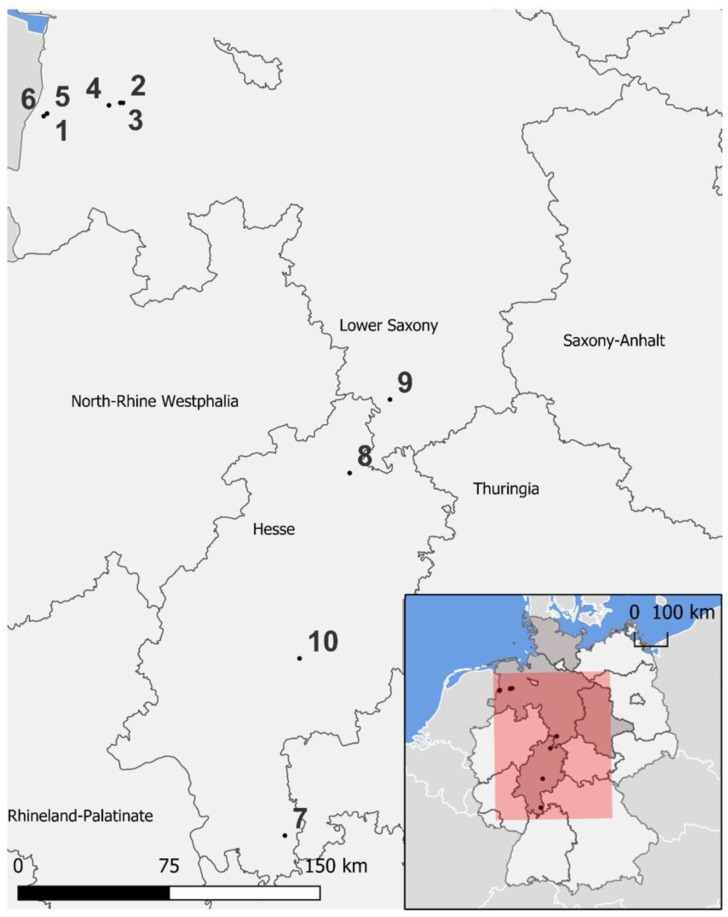
**Cases of disease;** the small map at the bottom right shows Germany with its federal states with locations of giant sequoia stands with cases of disease and neighbouring countries, the North Sea and Baltic Sea are highlighted in blue, the supporting federal states of the NW-FVA are highlighted in darker gray, the area marked in red is zoomed out in the large map and displays the locations of the ten analysed forest stands (1–10, Table 1) in Lower Saxony and Hesse. © GeoBasis-DE/BKG 2014 and © EuroGeographics.

**Table 1 pathogens-13-00715-t001:** Forest stand and site information for the disease cases of giant sequoia (*Sequoiadendron giganteum*) in Northwest Germany reported and analysed in the years 2018–2023.

Disease Case		Forest Site						*Sequoiadendron* *giganteum*	
No	Year of Report	m asl	Bed Rock	Average Annual Precipitation Sum (mm) ^1^	Deviation in 2018 from the Annual Average Precipitation ^1^	Average Precipitation Sum of the Vegetation Period (mm) ^1^	Precipitation Deficit in the Vegetation Period 2018 (%) ^1^	Tree Age (y)	Observed Symptoms (Figure 2)
1	2018	10	Sand	726	−22.4	388	39.7	Approx. 27	Dieback of the crown, shoot dieback, dead, brown leaves
2	2018	20	Sand	671	−29.7	355	60.2	12	
3	2018	20	Sand	612	+6.4	327	9.3	18	
4	2018	23	Sand	671	−29.7	355	60.2	25	
5	2018	10	Sand	671	−29.7	355	60.2	28	
6	2018	13	Sand	726	−22.4	388	39.7	21	
7	2021	172	Loess loam, which is anthropogenically modified in the urban area	820	−25.5	406	59.1	Approx. 60	Wood discolouration at the stem base along a tension crack
8	2021	212	Loam, which is anthropogenically modified in the urban area	606	−28.7	356	58.4	Approx. 12	Dieback of the crown (brown needles, shoots and branches) since 2020
9	2022	357	Buntsandstein	705	−24.7	370	51.1	Approx. 33	Dieback of the crown (brown needles, shoots and branches), wood discolouration and rot at the stem base and in branches
10	2023	300	Volcanic rocks of the Miocene	865	−21.3	420	45.7	Approx. 90	Dieback of the crown (brown needles, shoots and branches) since 2019

m asl: m above sea level. ^1^ According to data from adjacent DWD weather stations; for average data, the climate reference period 1961–1990 was used (Appendix A).

**Figure 2 pathogens-13-00715-f002:**
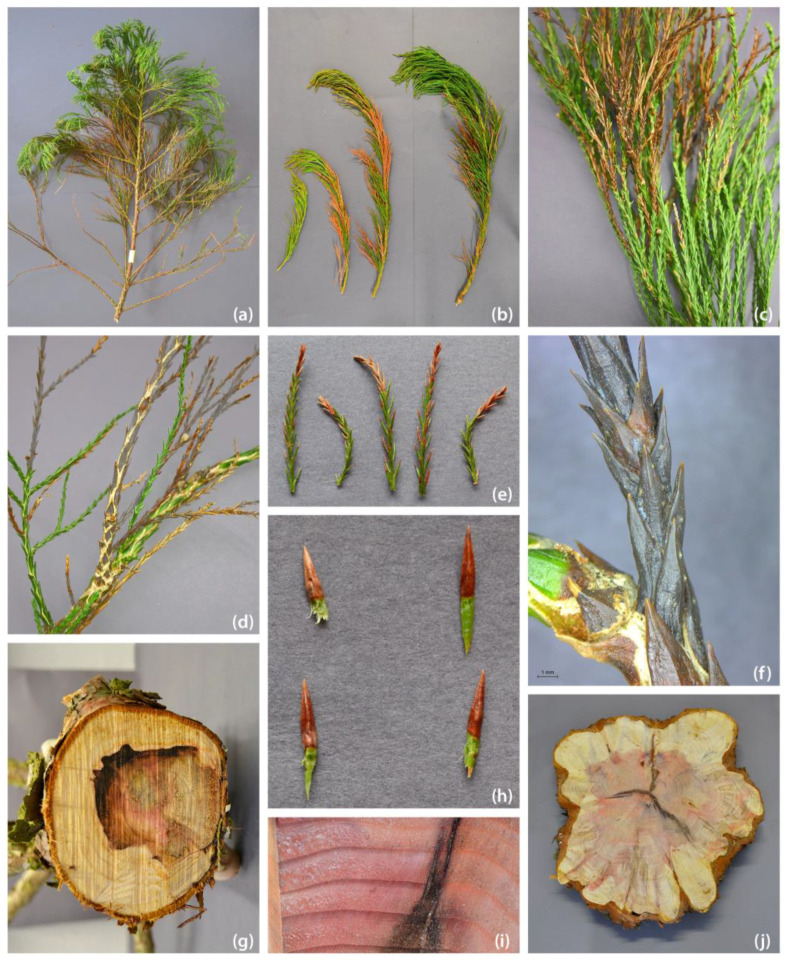
Various disease symptoms in giant sequoia. (**a**) Browning of shoots associated with *Amycosphaerella africana* and *Pestalotiopsis australis* in disease case 5; (**b**) browning of shoots associated with *Pestalotiopsis australis* and *Rhizosphaera minteri* in disease case 1; (**c**) browning of shoots where *Botryosphaeria dothidea* and *Neofusicoccum parvum* were isolated from disease case 10; (**d**–**f**) close-up of dying twigs from disease case 10; (**g**) wood discoloration; *Botryosphaeria dothidea* and *Pezicula neosporulosa* were isolated in disease case 9; (**h**) diseased needles of disease case 1; (**i**) close-up of wood discoloration associated with *Ophiostoma quercus* disease case 7; (**j**) stem disc from disease case 9; *Pezicula neospurolosa*, *Cantharellales* sp., and *Pezicula* sp. (*melanigena* or *radicicola*) were isolated.

### 2.2. Isolation of Fungi

Associated fungi of the giant sequoia samples were isolated from surfaces of sterilised woody chips obtained from the basal stem discs and stems according the method of Peters et al. (2023) [37]. Parts of the plant with visible degeneration were examined (browning, necrotisation; see Figure 2). Dieback pathogens were isolated according the method of Bußkamp et al. (2020) [38], with the difference that the shoots were not defoliated. Therefore, affected shoots were washed and surface-disinfested by treatment for 1 min in 70% EtOH, 5 min in 3% NaOCl, and 1 min in 70% EtOH. Affected needles were washed and surface-disinfested with treatment for 1 min in 70% EtOH, 1 min in a 3% NaOCl, and 1 min in 70% EtOH. Thereafter, twigs were cut into 5–6 segments with a size of 5 mm length, and needles were cut into three segments and then plated on malt yeast peptone agar (MYP). Emerging mycelia were subcultured separately on MYP medium. Isolated strains were first assigned to mycelial morphotypes (MT), which were further characterised based on micromorphological characteristics and DNA sequence analysis. Representative strains were stored in the fungal culture collection of the Northwest German Forest Research Institute (NW-FVA).

### 2.3. Detection of Heterobasidion annosum and Armillaria at the Stem Collar

The detection of *Heterobasidion annosum* was performed using the incubation method of Langer and Bressem (2017) [39] and the microscopic evaluation of fungal conidiophores produced on infected woody tissue. A Stemi 508 microscope (Zeiss, Jena, Germany) with an Axiocam ERc 5S and an Axiolab 5 microscope (Zeiss, Jena, Germany) with an Axiocam 208 color were used.

### 2.4. Identification of Isolated Fungi

At least one representative strain from each MT was chosen for molecular analysis. The genomic DNA of the isolates was extracted using the method of Damm et al. (2008) [40]. The 5.8S nuclear ribosomal gene with the two flanking internal transcribed spacers ITS-1 and ITS-2 (ITS region) was amplified for all strains. Additionally, for a selection of strains, the 28S nrRNA gene (LSU), a partial sequence of the translation elongation factor 1α (*EF-1α*), a partial DNA-directed RNA polymerase II second-largest subunit gene (*RPB2*), and a partial beta-tubulin gene (*TUB*) were amplified using the primer pairs and PCR conditions listed in Table 2. The PCR mixture consisted of 1 µL of DNA and 19 µL mastermix that contained 2.5 µL 10× PCR reaction buffer (with 20 mM MgCl_2_, Carl Roth, Karlsruhe, Germany), 1 µL of each primer (10 mmol), 2.5 µL MgCl_2_ (25 mmol), 0.1 µL RotiR-Pol Taq HY Taq polymerase (Carl Roth, Karlsruhe, Germany), and 2.5 µL of 2 mmol dNTPs (Biozym Scientific GmbH, Hessisch Oldendorf, Germany). Each reaction was topped up to a volume of 20 µL by adding sterile water. A StepOnePlus ™ PCR System (Applied Biosystems, Waltham, MA, USA) was used to carry out the DNA amplifications. A 1% agarose gel was used to visualise the PCR products. The products were sent to Eurofins Scientific Laboratory (Ebersberg, Germany) for sequencing. All resulting sequences were visually checked and edited as follows using BioEdit Sequence Alignment Editor (v. 7.2.5; Hall (1999) [41]). Consensus sequences were generated, improper sequence beginnings and ends were trimmed, and erroneous nucleotide allocations were corrected. Sequences were submitted to GenBank (Table 3, Appendix A).

Blastn searches using ITS sequences were carried out on the GenBank database (http://www.ncbi.nlm.nih.gov/genbank (accessed on 1 May 2024), Altschul et al. (1997) [42]) in order to determine the taxonomic classification of isolates. The results were critically interpreted with emphasis on well-curated culture collections, such as the Westerdijk Fungal Biodiversity Collection (CBS). For confirmation, the findings were rechecked against the literature and previously identified cultures from the institute’s collection. Extended analyses for taxon determination on a species level were conducted for isolates belonging to *Amycosphaerella*, *Coniochaeta*, *Phacidiopycnis*, and *Pestalotiopsis* (Figure 3, Appendix A). Phylogenetic analyses were conducted based on concatenated sequence datasets, including appropriate reference sequences retrieved from GenBank. All analyses were performed using RAxML v. 8.2.11 [43,44], as implemented in Geneious R11 [45], using the GTRGAMMA model with the rapid bootstrapping and search for best scoring ML tree algorithm including 1000 bootstrap replicates [46,47].

**Table 2 pathogens-13-00715-t002:** Primer pairs and PCR conditions used for molecular analyses of strains.

DNA-Region ^1^	Primer Pairs ^1^	PCR Conditions	Primer Reference
ITS	ITS-1F + ITS4	See Bien et al. (2020) [48]	Gardes & Bruns (1993) [49], White et al. (1990) [50]
LSU	LROR + LR5	See Paulin & Harrington (2000) [51]	Rehner & Samuels (1994), Vilgalys & Hester (1990) [46,47]
*RPB2*	RPB2-5F2 + RPB2-7cR	See Braun et al. (2018) for species of *Amycosphaerella* [52]; see Tanney & Seifert (2018) for species of *Phacidiopycnis* [53]	Liu et al. (1999) [54]
*EF-1α*	EF1-983F + EF1-2218R	See Arnold et al. (2021) for species of *Coniochaeta* [55]	Rehner & Buckley (2005) [56]
*TUB*	T1 + Bt-2b	See Liu et al. (2019) for species of *Pestalotiopsis* [54]	Glass & Donaldson (1995), O’Donnell & Cigelnik (1997) [57,58]

^1^ ITS: internal transcribed spacers and intervening 5.8S nrDNA; LSU: large subunit (28S) of the nrRNA gene operon; *RPB2*: partial DNA-directed RNA polymerase II second-largest subunit gene; *EF-1α*: partial translation elongation factor 1-alpha gene; *TUB*: partial beta-tubulin gene.

**Table 3 pathogens-13-00715-t003:** List of selected isolated fungi suggested to be noteworthy agents in the disease cases, alien to Germany, or new records for giant sequoia.

Species	Order	NW-FVA ID	Accession No	Disease Case	Shoot	Needle	Stem/Branch
*Amycosphaerella africana* ^1,2^	Mycosphaerellales	4336	PP913385	1–6	1	1	-
*Botryosphaeria dothidea*	Botryophaeriales	9830	PP913404	5, 9, 10	1	1	1
*Botryosphaeria parva* (anamorph: *Neofusicoccum parvum*) ^2^	Botryophaeriales	11986	PP913460	10	1	-	-
*Botrytis cinerea*	Helotiales	7912	PP913415	1–6, 8	1	1	-
*Coniochaeta acaciae* ^1,2^	Coniochaetales	9903	PP913418	9	1	-	-
*Coniochaeta velutina* ^1^	Coniochaetales	4360	PP913419	1	1	-	-
*Coprinellus micaceus*	Agaricales	9842	PP913421	9	1	1	1
*Diaporthe eres* A	Diaporthales	7949 11993	PP913426 PP913425	8, 10	1	1	-
*Diaporthe eres* B	Diaporthales	9864 7953 11989	PP913428, PP913427, PP913431	8–10	1	1	-
*Diaporthe nobilis*	Diaporthales	4447 4460 11994	PP913434 PP913433 PP913435	2, 3, 6, 10	1	1	-
*Diaporthe rudis*	Diaporthales	4359 9869	PP913436 PP913437	1, 9	1	1	-
*Muriformistrickeria rubi* ^1,2^	Pleosporales	12037	PP913455	10	1	-	-
*Nothophoma* cf. *quercina*	Pleosporales	7950	PP913464	8	-	1	-
*Ophiostoma quercus*	Microascales	6920	PP913465	7	-	-	1
*Pestalotiopsis australis* ^1,2^	Amphisphaeriales	4349 4341 4462 4497 4466	PP913472 PP913473 PP913474 PP913483 PP913475	1, 3, 5	1	1	-
*Pestalotiopsis monochaeta* ^1,2^	Amphisphaeriales	9837 9836 9832 11997	PP913476 PP913477 PP913478 PP913479	9, 10	1	1	1
*Pestalotiopsis* cf. *verruculosa*	Amphisphaeriales	7952	PP913480	8	-	1	-
*Pestalotiopsis* cf. *hollandica*	Amphisphaeriales	12038	PP913481	10	1	-	-
*Phacidiopycnis washingtonensis* ^1,2^	Rhytismatales	9901	PP913500	9	1		
*Phacidium lacerum*	Rhytismatales	4362 4449 4468	PP913501 PP913503 PP913502	1, 5, 6	1	-	-
*Phacidium* sp.	Rhytismatales	4488	PP913505	3	1	-	1
*Pseudocercospora* sp.	Mycosphaerellales	4456 4454	PP913506 PP913507	2	1	-	-
*Rhizosphaera minteri* ^1,2^	Venturiales	4358	PP913508	1	1		

^1^ New report for giant sequoia. ^2^ New report for Germany.

## 3. Results

### 3.1. Cases of Disease in Giant Sequoia in Northwest Germany

All the stands analysed suffered to varying degrees from precipitation deficits during the study period. With the exception of stand 3, all sites showed a deficit compared to the climate reference period 1961–1990 for the average annual precipitation sum in the period 2018 to 2023 (Table 1 and Appendix A). In 2018, all affected forest sites experienced a significant precipitation deficit (9.3–60.2%) during the vegetation period from May to October. Some locations also had a lack of rainfall in the following years (Appendix A). The observed disease symptoms on giant sequoia were dieback of the crown, shoot dieback, death, brown needles, and wood and rot discolouration at the stem (Figure 2).

### 3.2. Associated Fungi

In the studied woody tissues (Figure 2) of the stems, no *Heterobasidion annosum* nor *Armillaria* species were found. In total, 81 Dikaria species (Appendix A) were detected, of which 97.6% were Ascomycota and 2.4% Basidiomycota *(Cantharellales* sp. and *Coprinellus micaceus* (Bull.) Vilgalys, Hopple & Jacq. Johnson). The majority of the isolated species were assigned to the common endophytic fungal species that occur in conifers, according Bußkamp et al. (2020) and Langer et al. (2021) [38,59]. Isolated fungi that were new for Germany or giant sequoia or that were suggested to be noteworthy causative agents in the disease cases are listed in Table 3.

Fungal pathogens previously reported on giant sequoia in Germany, such as *B. dothidea* (*Botryosphaeriaceae*, Figure 3) and *Botrytis cinerea* Pers. (*Sclerotiniaceae*), were detected in the diseased tissues of cases 5, 9, 10 and 1–5, 8, respectively. *Botryosphaeria dothidea* was also retrieved from asymptomatic needle tissue (Appendix A). The isolated *Diaporthe* strains could be assigned, based on phylogenetic analysis, to the following species groups: *Diaporthe eres* Nitschke, *Diaporthe nobilis* Sacc. & Speg., and *Diaporthe rudis* (Fr.) Nitschke. Strains belonging to *D. eres* showed considerable differences in their ITS sequences and were, for the purpose of this study, subdivided into *D. eres* A and B.

Various strains of *Pestalotiopsis* (*Pestalotiopsidaceae*) were detected in the disease cases of 1–3, 5, 6, 8, 9, and 10. Based on the multigene phylogeny conducted, the strains isolated from giant sequoia can be distinguished into four taxa (Figure 3). Several strains showed concordance with the reference strains of *Pestalotiopsis australis* Maharachch., K.D. Hyde & Crous (Figure 4 and Figure 5e,f), including the ex-type strain. Despite the lack of bootstrap support of the clade, the isolated strains are presumed to belong to this species, due to considerable genetic differences to the closest related species *Pestalotiopsis scoparia* Maharachch., K.D. Hyde & Crous. A further four strains clustered in a well-supported clade with the ex-type strain of *Pestalotiopsis monochaeta* Maharachch., K.D. Hyde & Crous. One strain (NW-FVA 12038) clustered with several strains, including ex-type strains of *Pestalotiopsis hollandica* Maharachch., K.D. Hyde & Crous and *Pestalotiopsis brassicae* (Guba) Maharachch., K.D. Hyde & Crous. As *P. brassicae* has so far only been described from New Zealand [60], an assignment of the strain isolated here to *P. hollandica* seems appropriate. Due to the unclear genetic delimitation, the strain is referred to as *P.* cf. *hollandica*. One strain isolated here (NW-FVA 7952) groups with several strains tentatively assigned to *Pestalotiopsis verruculosa* Maharachch. & K.D. Hyde and the ex-type strain of this species. Due to missing support for the combined clade and phylogenetic similarity to other strains, including the ex-type strains of *P. brassicae* and *P. hollandica*, the strain is referred to as *P.* cf. *verruculosa*.

### 3.3. New Records for Giant Sequoia or Germany

Seven isolated species are new reports for Germany and giant sequoia (Table 3): *Amycosphaerella africana* (Crous & M.J. Wingf.) Quaedvl. & Crous (*Mycosphaerellaceae*, Figure 4 and Figure 5a,b)*, Coniochaeta acaciae* Samarak., Gafforov & K.D. Hyde (*Coniochaetaceae*, Figure 4 and Figure 5c,d), *Muriformistrickeria rubi* Q. Tian, Wanas., Camporesi & K.D. Hyde (*Melanommataceae*), *P. australis (*Figure 4 and Figure 5e,f)*, P. monochaeta*, *Phacidiopycnis washingtonensis* C.L. Xiao & J.D. Rogers (*Phacidiaceae*), and *Rhizosphaera minteri* Joanne E. Taylor & Koukol (*Venturiaceae*)*. Amycosphaerella africana* was associated with shoot and needle dieback in disease cases 1–6 in the north of Lower Saxony, but was also isolated from green shoots (disease case 4). *Coniochaeta acaciae* was only isolated with disease case 9 (located in the south of lower Saxony). From a symptomatic dead shoot of disease case 10 (located in Vogelsberg, Central Germany) we obtained a single isolate of *M. rubi. Pestalotiopsis australis* and *P. monochaeta* were isolated from several disease cases (1, 3, 5, and 9, 10, respectively). *Phacidiopycnis washingtonensis* was identified as a single isolate retrieved from giant sequoia shoots in disease case 9. *Rhizosphaera minteri* was isolated from eight out of ten sampled symptomatic or dead shoots in disease case 1.

*Neofusicoccum parvum* (disease case 10) was reported for the first time in Germany. The isolation of *Coniochaeta velutina* (Fuckel) Cooke from a dead symptomatic shoot of a giant sequoia (disease case 1) represents the first evidence of this fungal species on this host species. Additionally, a *Pseudocercospora* species (*Mycosphaerellaceae*) was isolated from shoots of disease case 2, showing a very high level of similarity in the ITS region with *Pseudocercospora lindericola* (W. Yamam.) Goh & W.H. Hsieh, *Pseudocercospora natalensis* Crous & T.A. Cout., and *Pseudocercospora rhododendri* Crous & Yuan Yuan Chen.

## 4. Discussion

### 4.1. Triggering Factors for Disease Outbreak

All studied diseased giant sequoia trees suffered from precipitation deficits during the study period and extreme summer drought and heat events characterised by climate anomalies [61,62]. This was, in the authors’ opinion, the main triggering factor for the observed outbreak of fungal diseases in giant sequoia. This is in accordance with the opinions of Choat et al. (2012) [63] and McDowell and Allen (2015) [64], who posit that drought and heat are key inciting factors in severe reductions in tree vigour in forests. Moreover, this is consistent with observations of the occurrence of other fungal forest tree diseases from 2018 to 2023 [1,5,65,66,67] and the assessment of the impact of the exceptional weather conditions between 2018 and 2022 on the health of our forest stands [65,66]. Severe droughts have caused widespread tree mortality in many forest biomes [66]. Tree mortality is a complex process that can be linked to multiple interacting abiotic and biotic factors [68]. Primarily, tree mortality is often physiologically based. This is evidenced by the failure of the plant hydraulic system, leading to extensive crown death and tree mortality during drought [66]. Drought has a significant secondary effect of weakening trees, making them more susceptible to secondary invaders and opportunistic pests, such as canker and root- and wood-rotting fungi, facilitating the transition of endophytes to pathogenic fungi [1,2,5].

### 4.2. Associated Fungi

As expected, numerous endophytes and latent pathogens previously reported from Germany were isolated from the analysed asymptomatic and symptomatic tissues of *Sequoiadendron giganteum*, for example *B. dothidea*, *B. cinerea*, *Diaporthe* spp., *Phacidium lacerum* Fr., and *Pestalotiopsis* spp. [25,69]. Branch or shoot damage to giant sequoia trees in Germany caused by or associated with *B. dothidea*, *Diaporthe* spp., *P. funerea*, or *B. cinerea* has already been observed [25].

*Botryosphaeria dothidea* was associated with most of the examined symptomatic tissues in this study, but was also present in asymptomatic needle tissue (disease case 9). The latter supports Smith et al.’s (1996) [70] findings that this species is a latent pathogen that also occurs endophytically in its host plants. As mentioned above, *B. dothidea* has already been described as a common causal agent of shoot dieback and Botryosphaeria canker on giant sequoia in Germany [25]. Additionally, *N. parvum* was found, which has been described as a cause of canker on giant sequoia in Switzerland [28]. According to the species list of the German red list on higher fungi, this is the first report of this species in Germany [69]. The latter *Botryophaeriaceae* species is considered to be an aggressive vascular pathogen and can cause severe dieback symptoms and death in its host plants [71,72,73]. In Switzerland, both *B. dothidea* and *B. parva* were found on diseased giant sequoia trees. Pathogenicity tests fulfilling the Henle–Koch postulates showed that *B. parva* was the main cause of the damage [28]. Both species are regarded as cosmopolitan fungi, and are associated with a variety of plant hosts, either as endophytes or as pathogens [74,75,76]. The authors hypothesise that *B. parvum* has previously been confused with *B. dothidea*, or that *B. parva* is a new pathogen on giant sequoia [29].

A *Pseudocercospora* species with a very high level of similarity in the ITS region with non-native *P. lindericola*, *P. natalensis*, and *P. rhododendri* (=*Chuppomyces handelii* (Bubák) U. Braun, C. Nakash., Videira & Crous) was isolated from a symptomatic shoot. The cosmopolitan genus *Pseudocercospora* comprises plant pathogenic fungi, has a wide range of host plants, and is generally associated with blights, as well as leaf and fruit spots [77]. *Pseudocercospora lichenum* (Keissl.) D. Hawksw. is the only species of this genus that is listed in the standard species list of fungi for Germany [69]. In contrast, *P. lindericola* was found on *Lauraceae* in Taiwan and China [78]. *Pseudocercospora natalensis* was originally described from *Eucalyptus nitens* (H.Deane & Maiden) Maid in South Africa [79] and *C. handelii* from *Rhododendron ponticum* L. in Turkey, but they have also been detected in the Netherlands [80].

### 4.3. New Records for Giant Sequoia or Germany

*Amycosphaerella africana* was frequently isolated from symptomatic needles and shoots of *S. giganteum* in this study. The species was originally described as *Mycosphaerella africana* Crous & M.J. Wingf. in 1996, after being isolated from leafspots of *Eucalyptus viminalis* Labill. in South Africa [81]. At least five synonyms are known since the species has been described multiple times under various names due to confusion concerning modes of ascospore germination [82]. According to Index Fungorum, current synonyms other than the basionym are *Mycosphaerella aurantia* A. Maxwell, *M. buckinghamiae* Crous & Summerell, *M. ellipsoidea* Crous & M.J. Wingf., and *Teratosphaeria africana* (Crous & M.J. Wingf.) Crous & U. Braun, all representing teleomorphic states. Additionally, *Uwebraunia ellipsoidea* Crous & M.J. Wingf. has been described as an anamorphic state for *M. ellipsoidea* (Crous & Wingfield 1996). *Amycosphaerella africana* has been primarily connected to leaf spot diseases of several *Eucalyptus* species [80], namely *Eucalyptus deanei* Maiden, *E. globulus* Labill., *E. grandis* W.Hill, *E. radiata* A.Cunn. ex DC., *E. smithii* F.Muell. ex R.T.Baker, and *E. viminalis*. Additionally, *A. africana* has been found to be connected to leaf spot disease in *Buckinghamia* sp. [83], and citrus greasy spot disease in four *Citrus* spp. [84]. It was found in leaves of *Fraxinus ornus* L. (endophytic occurrence) [85], *Metrosideros excelsa* Sol. ex Gaertn., and *Dracaena draco* (L.) L. [80]. The species appears to be widespread, with reports from Africa, Australia, South America, and New Zealand [81,86,87,88]. In Europe, it has been found in Italy, Portugal, and Spain [80,84,85,86,89]. This study presents the first report of *A. africana* from Germany, but more interestingly the first report of this species from a gymnosperm host. In addition, the location of this new find is considerably further north than all previous finds in warmer climates. The transmission and spread of mycosphaerella-like fungi, which cause leaf spot diseases, is thought to be linked to the increasing transportation of infected plant material between plantations in different countries or continents [90]. Temperature and moisture play a significant role in the establishment of pathogens in new environments [91]. Hence, the long periods of high temperatures in recent years could explain a new occurrence of *A. africana* in Germany.

The endophytic occurrence of *A. africana* in *F. ornus* in Northern Italy, a report from the monocotyledon plant *D. draco* in New Zealand, and the evidence presented here in *S. giganteum* in Northwest Germany suggest a much higher hidden distribution in possibly a large number of different host species. However, the question remains as to whether the fungal species was introduced to Germany in conjunction with the host, or whether the fungus jumped onto trees that were already present in Germany. The latter seems more likely in view of the fact that the natural ranges of the host and the fungus (as far as is known) do not appear to overlap. Considering the ability of the fungus to infect angiosperm and gymnosperm host species alike, further transmission within forest trees seems possible. As in other mycosphaerella-like fungi, transmission probably occurs through the wind dispersal of ascospores or the splash dispersal of conidia directly between host leaves [92,93,94]. In the case of leaf spot disease in *E. globulus*, Aguín et al. (2013) [89] suggest that *A. africana* acts as a primary pathogen, albeit with low incidence and aggressiveness, highlighting the possibility that the pathogen has been overlooked in forestry for a long time. Against this background, however, the pathogen does not appear to pose a major threat to German forests.

*Coniochaeta acaciae* was isolated once from an asymptomatic needle of giant sequoia in this study. This species was described in 2018 as a saprobic fungus from a dead trunk and branches of *Acacia* sp. in Uzbekistan [95]. Species of *Coniochaeta* have been isolated from a wide variety of substrates: saprobic, pathogenic, endophytic, lichen-associated, and even extremophilic lifestyles are known [96,97,98,99,100,101]. Apart from *Acacia* sp., *C. acaciae* has been reported from asymptomatic leaf petioles of *Fraxinus excelsior* L. in Poland [102], from dead twigs and branches of *Betula pendula* Roth in Ukraine [103], in connection to the lichen *Flavopunctelia flaventior* (Stirt.) Hale in Yunnan, China [104], and from *Anemone rivularis* Buch.-Ham. ex DC. in China [105]. According to an entry in Genbank, the species was further found in seeds of *Quercus ithaburensis* subsp. *macrolepis* (Kotschy) Hedge & Yalt. in Iraq (GenBank Acc. OQ185454). Based on the multigene (LSU-ITS-*EF1α*) phylogeny of *Coniochaeta* species in this study (Appendix A) and that of Si et al. (2021) [104], the closest relatives of *C. acacia* are saprobic *Coniochaeta baysunika* Wanas., Gafforov, E.B.G. Jones & K.D. Hyde from *Rosa* sp. in Uzbekistan [106], endophytic *C. euphorbiae* S. Nasr, S. Bien & Damm, and endophytic *C. iranica* S. Nasr, S. Bien & Damm from *Euphorbia polycaulis* Boiss. & Hohen. in Iran [107]. In this study, *C. acaciae* is reported for the first time from Germany and for the host *S. giganteum*. In the original description of the species, Samarakon et al. (2018) [108] describe perithecia with exceptionally short setae. Here, we report rather long setae, predominantly between 60 and 90 µm, sometimes up to 120 µm in length, from perithecia produced in culture after approximately 8 weeks, which unfortunately remained sterile (see Figure 5c,d). One further species of *Coniochaeta* was found in this study, namely *C. velutina*. This species is distributed worldwide and has already been reported in Germany [37,109,110,111,112]. To the authors’ knowledge, however, this is the first report on the host *S. giganteum*.

Here, we report *Muriformistrickeria rubi* Q. Tian, Wanas., Camporesi & K.D. Hyde for the first time from the new host *S. giganteum*, as well as from Germany. The two known members of the genus *Muriformistrickeria* are so far known as being hosted by the *Rosaceae* family. *Muriformistrickeria rubi* was described on a dead branch of *Rubus* sp. in Italy and has since been detected on spines of *Rosa* sp. in Sweden [106,113]. Its sister species, *Muriformistrickeria rosae* Wanas., Camporesi, E.B.G. Jones & K.D. Hyde, was also isolated in Italy from dead spines of *Rosa canina* L. [106]. No statement can be made about the pathogenic potential of these fungi due to the low number of findings and the lack of information concerning their ecological roles.

*Phacidiopycnis washingtonensis* was found to be associated with the shoot dieback of *S. giganteum* in this study. This anamorphic species was described in 2005 as a post-harvest pathogen of apples (*Rosaceae*) in Washington, USA [114]. It is the causal agent for the disease named “speck rot” or “rubbery rot” on apple fruit and persimmon fruit of *Diospyros kaki* Thunb. [115,116,117]. Additionally, the fungus has also been isolated in connection with canker disease and twig dieback on crabapple and pear trees in commercial orchards [114] and with a leaf blight disease on *Arbutus menziesii* Pursh [118]. In the fruit industry, this fungal infection possesses the capability to induce significant post-harvest damage [114,116] and has been reported from the Northwest USA, Northern Italy, Northern Germany, Chile, and Norway [114,115,117,119,120]. This study presents the first report of *P. washingtonensis* from giant sequoia and at the same time the first report from a gymnosperm host and a forest environment in Europe. Since the pathogen has been widely reported in the Northwestern United States and is thus distributed within the natural range of *Sequoiadendron*, an introduction into Germany together with the host seems probable, in contrast to *A. africana*, as discussed above. Weber (2011) [117] does not rule out a recent migration of the fungus to Europe, but considers a longer, hidden spread of the pathogen in fruit orchards to be more likely. In the past, infections in German fruit orchards or post-harvest might have remained unnoticed or ignored due to marginal damages [117]. Additionally, the recognition of this specific fruit rot disease is hindered due to simple confusion with diseases causing similar symptoms, such as infection with *B. cinerea*, *Sphaeropsis*, *Phytophthora*, or *Monilia* [116,117]. In the case of *P. washingtonensis*, the fungus does not seem to benefit from rising temperatures. Xiao et al. (2005) [114] describe the fungus as a low-temperature species that grows between −3 °C and +25 °C and finds its optimum growth between 15 and 20 °C. Above 30 °C, the fungus reversibly stops growing; according to the study, temperatures of +35 °C over a period of 10 days led to the death of cultures. The observations by Elliott et al. (2014) [118] led to the assumption that foliar blight on *A. menziesii* is triggered when the leaves are subjected to cold stress. Increased disease severity in the spring of 2011 in the Northwestern USA was linked to extreme cold in the preceding months of November and February. Although cold extremes in Germany cannot be ruled out, long periods of heat and drought, such as those that have increasingly occurred in the study area in recent years, would probably prevent a dangerous establishment of the pathogen. Accordingly, an increase in abnormalities associated with this fungus in German forests is not to be expected. However, a long-hidden distribution of *P. washingtonensis* in German forests seems possible in view of the fact that a closely related fungal species is known in the wood and needles of conifer hosts in Northern Europe. Multilocus phylogeny shows a close relationship to *Allantophomopsiella pseudotsugae* (M. Wilson) Crous (Appendix A), which has been isolated from *Abies*, *Larix*, *Pinus*, *Picea*, *Pseudotsuga*, and *S. giganteum*, also in Germany [25,121,122]. It can be assumed that a risk from *P. washingtonensis* exists for trees belonging to the *Rosaceae* family, based on the occurrences described above. However, according to Amiri (2020) [123], *P. washingtonensis* is considered as a weak canker- and twig-dieback-causing pathogen on trees.

Anyway, we did not isolate *A. pseudotsugae*. On the contrary, we found two other *Phacidiaceae*, namely *P. lacerum* (=*Dothidea pinastri* Fr., =*Ceuthospora pinastri* (Fr.) Höhn. fide Crous et al. (2014) [121]) and *Phacidium* sp. *Phacidium lacerum* is a fungus that has been found throughout Europe, including Germany, and was first described from *Pinus sylvestris* L. needles [121]. Therefore, this species which has a wide host range within pine species [124,125], is considered by the authors of this study to be native to Germany. An additional *Phacidium* species, *Phacidium pseudophacidioides* Crous, is native to Europe and occurs in the Netherlands, sampled from *Ilex aquifolium* L. and in Switzerland, sampled from *Chamaespartium sagittale* (L.) P.E.Gibbs [121]. It is therefore likely that the latter species could be native to Germany but has not yet been found or has been assigned to *P. lacerum.*

Only two of the isolated *Pestalotiopsis* species, *P. australis* and *P. monochaeta*, could be clearly identified down to species level. *Pestalotiopsis funerea*, which can occur as a concomitant pathogen on various tree species [126,127,128,129] and is said to occur on giant sequoia in Germany [25], has not been identified. However, the inability to recognise *P. funerea* might be explained through the lack of reliable sequence data of this species [130,131]. The sequence data of two strains designated as *P. funerea* retrieved from GenBank were included in the phylogeny presented here (Figure 3), and are clearly distinct. The results of this study are the first published evidence for *P. australis* in Germany and on giant sequoia. *Pestalotiopsis australis* was originally described from *Telopea* sp. in Australia [60]. However, the species has recently been detected in Europe (Portugal) from diseased shoots of *Pinus pinea* L. and from diseased blueberries [131,132]. Infection experiments have shown that *P. australis* can cause lesions on blueberries, but not on *P. pinea.* Infection trials, as conducted by Heanzi et al. (2021) and Kovac et al. (2021) [28,32] on young giant sequoia trees, could be used to observe the influence of *Pestalotiopsis* species on the damage progress. Morphological studies on cultural characteristics and conidial morphology (e.g., Ref. [60]) will clarify the exact species affiliations, especially in the cases of *P. hollandica* and *P. monochaeta*. Judith-Hertz (2016) [108] suggested the synonymisation of *P. hollandica* and *P. monochaeta* based on genetic data. However, considerable morphological differences have been described for the two supposed species, such as the number of setulae (apical appendages) on the conidia (*P. hollandica* with 1–4 apical appendages, *P. monochaeta* with 1 apical appendage) and slight differences in culture morphology [60]. In the phylogeny presented here, the two ex-type strains can clearly be distinguished through the comparison of *TUB* (5 nk difference) and *EF-1α* (2 nk difference) sequences. *Pestalotiopsis hollandica* was first isolated from *Sciadopitys verticillata* (Thunb.) Siebold & Zucc. (*Sciadopityaceae*) in the Netherlands in 1933 [60]. This species has also been associated with *P. pinea* (*Pinaceae*), which showed shoot blight disease mainly caused by *Pestalotiopsis pini* A.C. Silva, E. Diogo & H. Bragança in Portugal [131]. Additionally, it has been identified in *Cupressus sempervirens* L. (*Cupressaceae*) in Spain [133] and in *Quercus robur* L. (*Fagaceae*) in the Netherlands [134]. The findings in different regions of Europe since 1933 suggest that *P. hollandica* could be native to Germany. *Pestalotiopsis monochaeta* was first described from *Quercus robur* L. and *Taxus baccata* L. in the Netherlands [60]. No other finds of this species have been published to date. There is some evidence that the species occurs in Germany (reported as *Pestalotiopsis* (cf.) *monochaeta*) [135].

*Rhizosphaera minteri* has recently been described from needles of *Picea abies* (L.) H. Karst. and *Picea sitchensis* (Bong.) Carrière in Scotland and Wales [136]. The isolates of this species in this study, from several browning shoots of one tree, represent the first report of the species on *Sequoiadendron* and outside Great Britain. Taylor et al. (2023) [136] emphasise that the species could only be found on dead needles, and it can therefore be assumed that it is a true needle-cast pathogen. Furthermore, the authors assume that the species has its natural range in North America and has migrated to Europe together with spruce hosts. The find shown here reveals another possible host with which the migration from North America could have taken place. As shown by Taylor et al. (2023) [136], *R. minteri* is closely related to *Rhizosphaera kalkhoffii Bubák* based on ITS phylogeny. The latter has already been associated with a host from the *Sequoioideae* group, as it has been found on *Sequoia sempervirens* (D.Don) Endl. in New Zealand [137,138]. The actual distribution and danger of *R. minteri* as a pathogen on gymnosperm hosts must be further investigated, as confusion with the morphologically similar *Rhizosphaera pini* (Corda) Maubl. is likely [136].

## 5. Conclusions

The first main objective of this study was fulfilled by isolating 81 Dikaria species from diseased tissues of giant sequoia. The second main objective was also partially achieved, as several fungal species associated with the diseased studied trees were isolated from giant sequoias for the first time or were new to Germany. It is still unclear whether these pathogens are truly invasive or non-native in Germany and whether they caused the observed damage. Additionally, the location of the first spatial contact between giant sequoia trees and these pathogens remains unknown. *Phacidiopycnis washingtonensis* could have been brought by the giant sequoia from its native range, as this fungus was first described in North America [114]. The third main objective was also realised, in which the authors assessed and discussed the risk to native tree species from the fungal species isolated that were assigned as alien pathogens in Germany. None of the pathogens was found to be a lethal causal agent on the giant sequoia trees studied in the absence of co-occurring other fungal pathogens and inciting abiotic factors. Although some of the fungal species reported for the first time in this study were detected in more than one disease case (*A. africana*, *P. australis*, *P. monochaeta*), the majority were restricted to a single case or only to a low number of tissue samples. This indicates a rather rare occurrence of the pathogens in relation to giant sequoia. However, the invasive behaviour of newly migrated species can be delayed by lag phases that can last for decades [139]. In conclusion, the cultivation of non-native giant sequoia can pose a potential risk to other native tree species in Germany, as the potentially alien fungi found could infect the latter and lead to novel diseases. It is therefore advisable to carry out pathogenicity tests with giant sequoia and native main tree species with *A. africana*, *C. acaciae*, *M. rubi*, *Pestalotiopsis* spp., *P. washingtonensis*, and *Phacidium* sp. in order to determine Koch’s postulates, in line with Bhunjun et al. (2021) [140].

We have not been able to prove conclusively whether the cultivation of non-native tree species in Germany is a gateway for alien species. It is not clear whether the newly discovered fungi originate from imported giant sequoia or whether the trees tested were infected here in Germany. It is also possible that the fungi found for the first time in Germany were previously undetected. However, the example of giant sequoia has shown that many fungal species unknown in Germany, including tree pathogens, have been discovered for the first time, and the risk should not be underestimated.

## Figures and Tables

**Figure 3 pathogens-13-00715-f003:**
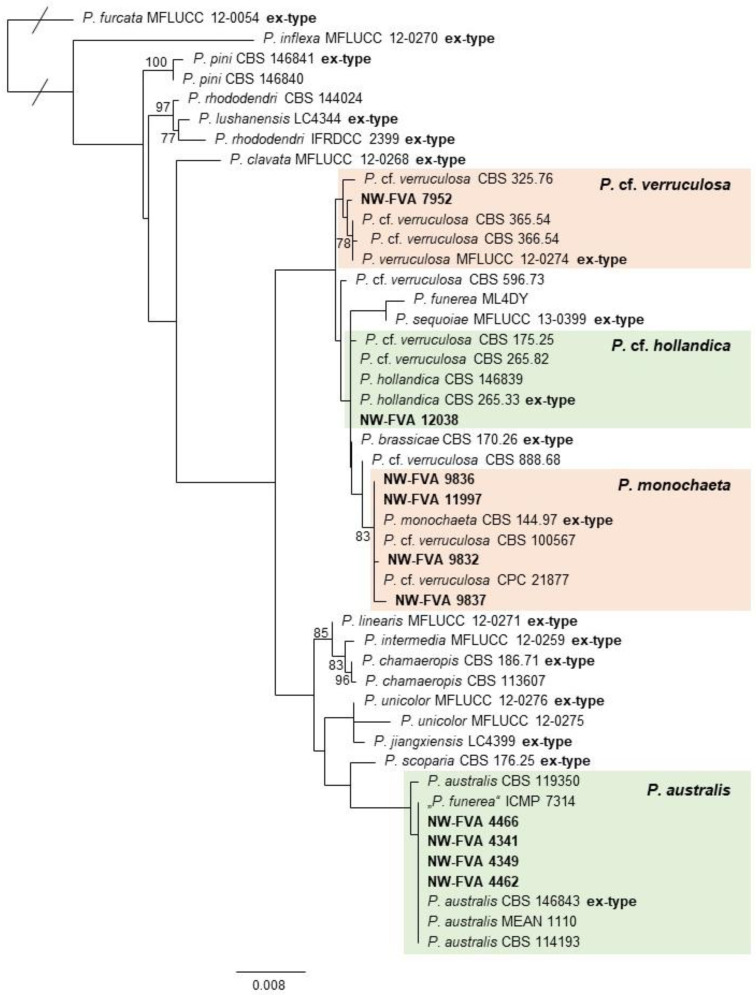
Phylogeny obtained by maximum likelihood analysis of the combined LSU-ITS-*TUB*-*EF1α* sequence alignment of species from *Pestalotiopsis*. ML bootstrap support values above 70% are shown at the nodes. *Pestalotiopsis furcata* strain MFLUCC12-0054 is used as the outgroup. Strains analysed in this study are emphasised in bold. Branches that are crossed by diagonal lines are shortened by 50%.

**Figure 4 pathogens-13-00715-f004:**
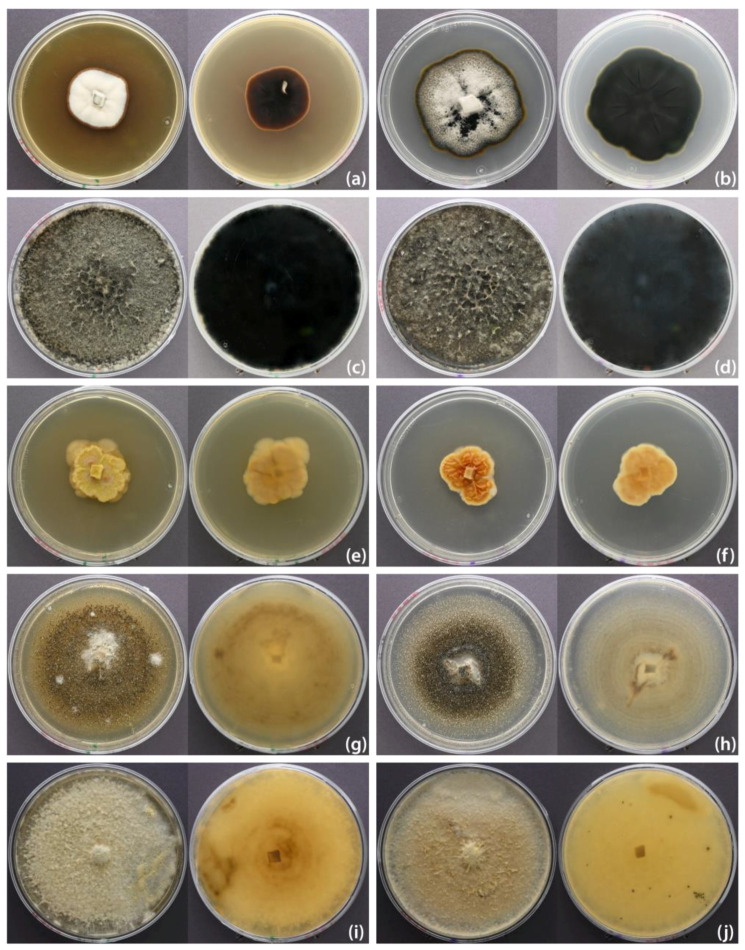
Colony surfaces of isolated fungal species on MEA (**left**) and PDA (**right**) medium after 4 weeks. (**a**,**b**) *Amycosphaerella africana* strain NW-FVA 4336; (**c**,**d**) *Botryosphaeria dothidea* strain NW-FVA 9830; (**e**,**f**) *Coniochaeta acaciae* strain NW-FVA 9903; (**g**,**h**) *Ophiostoma quercus* strain NW-FVA 6920; (**i**,**j**) *Pestalotiopsis australis* strain NW-FVA 4349.

**Figure 5 pathogens-13-00715-f005:**
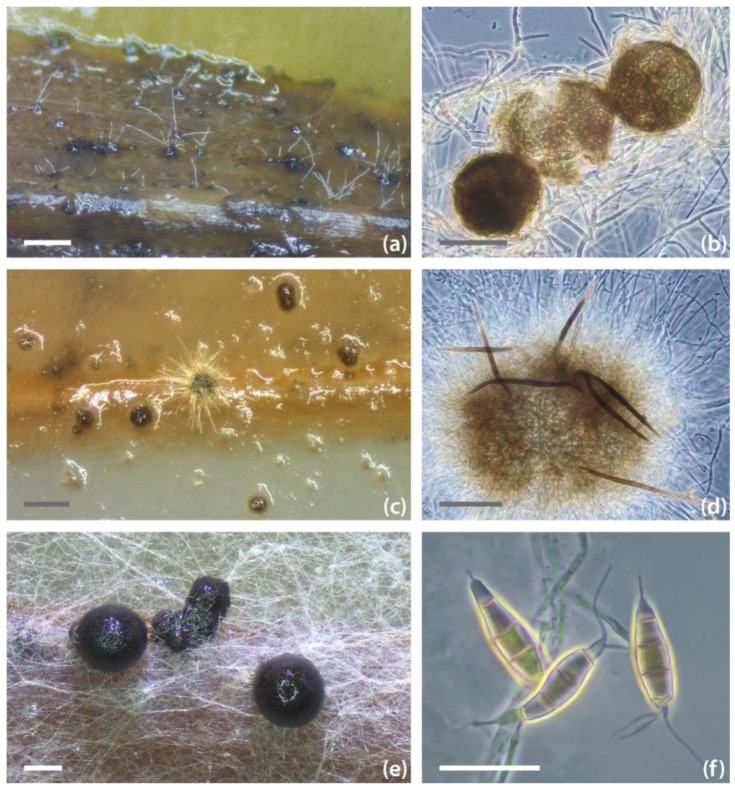
Microscopic illustrations of selected fungal species. (**a**,**b**) Conidiomata of *Amycosphaerella africana* strain NW-FVA 4336; (**c**,**d**) conidiomata of *Coniochaeta acaciae* strain NW-FVA 9903; (**e**) conidiomata and (**f**) conidia of *Pestalotiopsis australis* strain NW-FVA 4349. Scale bars: (**a**,**c**) = 200 μm; (**b**,**d**) = 50 μm; (**e**) = 128 μm; (**f**) = 20 μm.

## Data Availability

Sequence data of all sequences generated in this study (Table 3 and Appendix A) are deposited in the NCBI GenBank database.

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
