# Peer review of "Filamentous Fungi Associated with Disease Symptoms in Non-Native Giant Sequoia (Sequoiadendron giganteum) in Germany—A Gateway for Alien Fungal Pathogens?"

_pathogens, 2024, doi:10.3390/pathogens13090715_

Round 1
Reviewer 1 Report
Comments and Suggestions for Authors
This manuscript isolated 81 Dikaria species from diseased tissues of giant sequoia cultured in Germany. With a series of analysis, the authors speculate that filamentous fungi associated with disease symptoms on non-native giant sequoia in Germany might be a gateway for alien fungal pathogens. The data analysis is detailed and the discussion is sufficient.
1. Figure 2, the caption lacks an introduction to (H).
2. L141, “m asl m above see level.” Here, “see level” means “sea level”?
3. Table 3, the names of Order should be in regular font, not italics.
Author Response
Thank you very much for taking the time to review this manuscript. Please find the detailed responses below and the corresponding revisions/corrections highlighted in track changes in the re-submitted files.
Answers
Comments and Suggestions for Authors from 1st reviewer
This manuscript isolated 81 Dikaria species from diseased tissues of giant sequoia cultured in Germany. With a series of analysis, the authors speculate that filamentous fungi associated with disease symptoms on non-native giant sequoia in Germany might be a gateway for alien fungal pathogens. The data analysis is detailed and the discussion is sufficient.
- Figure 2, the caption lacks an introduction to (H).
Answer 1: done as follows:
Figure 2. Various disease symptoms on giant sequoia. (a) browning of shoots associated with Amycosphaerella africana and Pestalotiopsis australis in disease case 5; (b) browning of shoots associated with Pestalotiopsis australis and Rhizosphaera minteri in disease case 1; (c) browning of shoots where Botryosphaeria dothidea and Neofusicoccum parvum were isolated from disease case 10; (d, e, f) close-up of dying twigs from disease case 10; (g) wood discoloration, isolated were B. dothidea and Pezicula neosporulosa in disease case 9; (h) diseased needles of disease case 1; (i) close-up of wood discoloration associated with Ophiostoma quercus disease case 7; (j) Stem disc from disease case 9, isolated were Pezicula neospurolosa, Cantharellales sp., and Pezicula sp. (melanigena or radicicola).
- L141, “m asl m above see level.” Here, “see level” means “sea level”?
Answer 2: done
- Table 3, the names of Order should be in regular font, not italics.
Answer 3: done
Reviewer 2 Report
Comments and Suggestions for Authors
Introduction part is lengthy, kindly reduce it and ensure smooth transitions between paragraphs and maintain a flow of information.
Italicize species names for scientific accuracy throughout the manuscript and mention full name when use first and abbreviate it after (Sequoiadendron giganteum............S. giganteum).
Figure 4. Colony surface of isolated fungal species on MEA (left) and PDA (right) medium after 4 300 weeks. (a, b) A. africana strain NW-FVA 4336; (c, d) B. dothidea strain NW-301 FVA 9830; (e, f) C. acaciae strain NW-FVA 9903; (g, h) O. quercus strain NW-302 FVA 6920; (i, j) P. australis strain NW-FVA 4349.
pg14, lin3 360: Pseudocercospora lindericola occurs on Lauraceae in Taiwan and China.....
no need to write much info of spp like (Pseudocercospora lichenum (Keissl.) D. Hawksw.)....only P. lichenum is enough.
pg14, line 349-364: at some place you mentioned Pseudocercospora, at some you mentioned it with P, while Ps. kindle use similar and better to mention P. sp. Check this mistake throughout the manuscript and correct it with all sp name.
By breaking down the information into clear, manageable sections and maintaining consistency in formatting and terminology, this manuscript will be more engaging and accessible to readers.
I found too much irrelevant info in the manuscript which increased the references too, kindly rewrite the manuscript and add only related topic information.
Author Response
Thank you very much for taking the time to review this manuscript. Please find the detailed responses below and the corresponding revisions/corrections highlighted in track changes in the re-submitted files.
Comments and Suggestions for Authors from 2nd reviewer
- Introduction part is lengthy, kindly reduce it and ensure smooth transitions between paragraphs and maintain a flow of information.
Answer 1: done
We have reduced the introduction.
Now, the improved manuscript consist of 140 references: 82637 characters including blanks, 12154 words.
We also added new subheadings in results and discussion.
- Italicize species names for scientific accuracy throughout the manuscript and mention full name when use first and abbreviate it after (Sequoiadendron giganteum............S. giganteum).
Answer 2:
We italicize species names for scientific accuracy as a rule.
Authors have already done this for S. giganteum as a rule when mentioned first time in the manuscript apart from title and abstract, for example:
Introduction paragraph 4:
The giant sequoia (Sequoiadendron giganteum (Lindl.) J.Buchholz) in the family of Cupressaceae originates from California (Sierra Nevada). There it is prevalent in a highly disjunct area consisting of ~70 groves stretching about 400 km from north to south. Due to an expected decrease in soil moisture throughout the Sierra Nevada as a result of climate change, S. giganteum is threatened [23].
But as is good scientific practice, we have written the scientific names in full at the beginning of a sentence, or in legends, in listings, in new sections and in main conclusions.
But in the case of e.g. Botryosphaeria dothidea (Moug.) Ces. & De Not. and in some other case the reviewer was correct.
As recommended we use full names also in listings and also use the abbreviations in the latter sections and paragraphs.
In the case of Pestalotiopsis we used the abbreviation Pe. Instead of P. so that there are no misunderstandings with Pseudocercospora, Phacidium and Phacidiopycnis and their abbreviations P. But we changed all abbreviations of Pestalotiopsis and the other genera to P.
- Figure 4. Colony surface of isolated fungal species on MEA (left) and PDA (right) medium after 4 300 weeks. (a, b) A. africana strain NW-FVA 4336; (c, d) B. dothidea strain NW-301 FVA 9830; (e, f) C. acaciae strain NW-FVA 9903; (g, h) O. quercus strain NW-302 FVA 6920; (i, j) P. australis strain NW-FVA 4349.
Answer 3:
Original:
Figure 4. Colony surface of isolated fungal species on MEA (left) and PDA (right) medium after 4 weeks. (a, b) Amycosphaerella africana strain NW-FVA 4336; (c, d) Botryosphaeria dothidea strain NW-FVA 9830; (e, f) Coniochaeta acaciae strain NW-FVA 9903; (g, h) Ophiostoma quercus strain NW-FVA 6920; (i, j) Pestalotiopsis australis strain NW-FVA 4349.
If it is in accordance the Journal guidelines we can change it to:
Figure 4. Colony surface of isolated fungal species on MEA (left) and PDA (right) medium after 4 weeks. (a, b) A. africana strain NW-FVA 4336; (c, d) B. dothidea strain NW-FVA 9830; (e, f) C. acaciae strain NW-FVA 9903; (g, h) O. quercus strain NW-FVA 6920; (i, j) P. australis strain NW-FVA 4349.
However, the authors have learnt that it is standard practice that an illustration together with its legend must be understood independently of the manuscript text, so the species names have not been abbreviated.
- pg14, lin3 360: Pseudocercospora lindericola occurs on Lauraceae in Taiwan and China.....
Answer 4: done: In contrast, P. lindericola occurred on Lauraceae in Taiwan and China
- no need to write much info of spp like (Pseudocercospora lichenum (Keissl.) D. Hawksw.)....only P. lichenum is enough.
Answer 5: done: we have removed the additional information
- pg14, line 349-364: at some place you mentioned Pseudocercospora, at some you mentioned it with P, while Ps. kindle use similar and better to mention P. sp. Check this mistake throughout the manuscript and correct it with all sp name.
Answer 6: done
- By breaking down the information into clear, manageable sections and maintaining consistency in formatting and terminology, this manuscript will be more engaging and accessible to readers.
Answer 7: The authors have endeavoured to write the manuscript in a uniform and well-structured manner. In particular, material and methods are organised with numbered subheadings. This was also improved for the results and discussion if appropriate. However, as so many individual species are discussed, it was decided not to formulate a separate heading or paragraph for each one.
e.g.
3.1. Cases of disease on giant sequoia in Northwest Germany
3.2. Associated fungi
3.3. New records for giant sequoia or Germany
4.1. Triggering factors for disease outbreak
4.2. Associated fungi
4.3. New records for giant sequoia or Germany
We are not aware of the gross, consistent formatting errors or inconsistencies we have made in the terminology. It would have been nice to have had an example of this, apart from the naming and abbreviation of scientific species names.
8) I found too much irrelevant info in the manuscript which increased the references too, kindly rewrite the manuscript and add only related topic information.
Answer 8: As forest pathologists and mycologists, we found the information presented to be very useful in assessing the risk posed by the species identified.
But we have shortened it in accordance with the recommendations and reduced the cited references.
Round 2
Reviewer 2 Report
Comments and Suggestions for Authors
Table3:
Botryosphaeria parva..........B. parva
Coniochaeta acaciae should be like C. acaciae.
Coniochaeta velutina.......C. velutina.
And follow the same for all others.